# Differences in Hemodynamic Alteration between Atherosclerotic Occlusive Lesions and Moyamoya Disease: A Quantitative ^15^O-PET Study

**DOI:** 10.3390/diagnostics11101820

**Published:** 2021-10-01

**Authors:** Chiaki Igarashi, Hidehiko Okazawa, Muhammad M. Islam, Tetsuya Tsujikawa, Toshifumi Higashino, Makoto Isozaki, Ken-ichiro Kikuta

**Affiliations:** 1Department of Neurosurgery, Faculty of Medical Sciences, University of Fukui, Eiheiji-cho 910-1193, Japan; chiakii@u-fukui.ac.jp (C.I.); maruhiga@u-fukui.ac.jp (T.H.); isoiso@u-fukui.ac.jp (M.I.); kikuta@u-fukui.ac.jp (K.-i.K.); 2Biomedical Imaging Research Center, University of Fukui, Eiheiji-cho 910-1193, Japan; mmi@bme.kuet.ac.bd (M.M.I.); awaji@u-fukui.ac.jp (T.T.); 3Department of Biomedical Engineering, Khulna University of Engineering & Technology, Khulna 9203, Bangladesh

**Keywords:** cerebral hemodynamics, ^15^O-PET, arterial steno-occlusive lesion, moyamoya disease, cerebral perfusion pressure

## Abstract

To clarify the differences in hemodynamic status between atherosclerotic steno-occlusive lesions (SOL) and moyamoaya disease (MMD), hemodynamic parameters were compared using ^15^O-PET. Twenty-four patients with unilateral SOL (67 ± 11 y) and eighteen with MMD (33 ± 16 y) were assigned to this study. MMD patients were divided into twelve unilateral and six bilateral lesions. All patients underwent ^15^O-PET to measure cerebral blood flow (CBF), blood volume (CBV), oxygen extraction fraction (OEF), and metabolic rate (CMRO_2_). Acetazolamide was administered after the baseline scan and the second ^15^O-water PET was performed to evaluate cerebrovascular reactivity (CVR). For the CBF calculation in ^15^O-water PET, the three-weighted integral method was applied based on a one-tissue compartment model with pixel-by-pixel delay correction to measure precise CBF and arterial-to-capillary blood volume (V_0_). Baseline hemodynamic parameters showed significantly lower CBF, V_0_, and CMRO_2_, but greater CBV, OEF, and delay (*p* < 0.01) in the affected hemispheres than in the unaffected hemispheres. After ACZ administration, both hemispheres showed a significant increase in CBF (*p* < 0.0001), but not in V_0_. CVR differed significantly between the hemispheres. The arterial perfusion pressure of the functioning arterial part tended to be reduced after acetazolamide administration in patients with past neurologic events caused by hemodynamic impairment. MMD patients showed greater inactive vascular and venous volumes compared with common atherosclerotic SOL patients. The hemodynamic status of cerebral circulation may vary according to the chronic process of steno-occlusive change and the development of collateral circulation. In order to evaluate physiologic differences between the two diseases, ^15^O-PET with an acetazolamide challenge test is useful.

## 1. Introduction

Atherosclerotic steno-occlusive lesions (SOL) in the internal carotid artery (ICA) or the middle cerebral artery (MCA) cause approximately 10% of transient ischemic attacks (TIAs) and 15–25% of ischemic strokes in the cerebral carotid artery territory [1,2,3]. Moyamoya disease (MMD), characterized by progressive stenosis of the terminal portion of the ICA with moyamoya vessels, i.e., an abnormal vascular network in the anterior cerebral artery (ACA) and MCA, causes 3.2% of all strokes. Although the etiology of MMD is controversial, past intracranial infections of the patient may be one of the causes associated with chronic occlusive changes [4]. MMD progression is associated with ischemic events or silent infarctions in four of five patients [5]. For these patients with cerebrovascular disease (CVD), hemodynamic evaluation is important to assess the necessity for neurosurgical treatment including extracranial-intracranial bypass surgery due to their high rate of ischemic events and recurrent strokes [6]. Hemodynamic status caused by the reduction in cerebral perfusion pressure due to a chronic cerebral arterial occlusive change is usually classified into three stages: stage 0, normal condition; stage I, vasodilatory change with an increase in cerebral blood volume (CBV) without changes in cerebral blood flow (CBF) and oxygen extraction fraction (OEF); and stage II, CBF decline and OEF elevation without a reduction in the cerebral metabolic rate of oxygen (CMRO_2_) [7,8]. These stages are commonly used to determine treatment options for SOL patients. Although MMD is usually associated with increased CBV and OEF, and decreased CVR due to chronic cerebral vasodilation in the affected area, similar to SOL [9], the precise mechanisms of hemodynamic changes in MMD still remain unknown.

Positron emission tomography (PET) is a commonly used tool that allows for the accurate quantitative measurement of the hemodynamic status of both cerebral circulation and oxygen metabolism in CVD patients [10,11]. The recent development of CBF calculation based on a one-tissue compartment model (1-TCM) in ^15^O-water PET studies allows the measurement of small vascular volumes (V_0_) representing arterioles to the arterial part of capillaries [11,12]. The pixel-by-pixel delay correction in the method enables a better and more precise estimation of regional CBF and V_0_ values as well as dynamic changes in hemodynamics compared with the conventional approach [11,13]. This new approach is expected to delineate the different hemodynamic statuses in various stages of CVD, which may reveal different mechanisms of physiological compensation between SOL and MMD. The precise hemodynamic evaluation may facilitate the appropriate treatment of CVD depending on the stage and degree of impaired circulation. The purpose of this study was to clarify the differences in cerebral hemodynamics between atherosclerotic SOL and MMD by comparing the hemodynamic parameters obtained by our new pixel-by-pixel calculation method based on a 1-TCM [13].

## 2. Materials and Methods

### 2.1. Patients

Twenty-four patients with unilateral major cerebral arterial SOL (twenty men and four women; age 37–84, mean 67 ± 11 years old) and eighteen patients with MMD (seven men and eleven women; age 12–70, mean 33 ± 16 years old) were assigned to this study (during the period from January 2011 to December 2014) to evaluate cerebral hemodynamic changes using ^15^O-PET. The location of arterial lesions was determined by magnetic resonance imaging (MRI) and MR angiography (MRA) (Table 1). Patients with bilateral SOLs without moyamoya vessels were excluded. Patients who had undergone neurosurgical treatment were also excluded. Ten patients had right-side and fourteen had left-side SOLs with significant stenosis (*n* = 12; diameter reduction > 90%) or occlusion (*n* = 12) of the internal carotid artery (ICA) (*n* = 17) or the middle cerebral artery (MCA) (*n* = 7). Twelve MMD patients had unilateral steno-occlusive changes (right: *n* = 6, left: *n* = 6), and six had bilateral lesions. The lesioned side of MMD was determined by the results of MRA and/or angiography before the ^15^O-PET studies. Two SOL patients and one MMD patient had cortical infarctions in the parietal lobes. Other patients did not have cortical infarctions except for small lacunar infarctions or chronic ischemic changes in the white matter. Eleven patients did not have any lesions other than steno-occlusive changes in the major cerebral arteries. The study was approved by the Ethics Committee of the University of Fukui Hospital based on its guidelines (Ethical Guidelines for Medical Science Research with Humans) as well as the Helsinki Declaration of 1975 (revised in 1983). Written informed consent was obtained from each patient.

### 2.2. PET Study

All patients underwent ^15^O-gas and water PET with a whole-body PET scanner (Advance, GE Medical Systems, Milwaukee, WI, USA). The PET scan permits simultaneous acquisition of 35 image slices in a two-dimensional mode with an interslice spacing of 4.25 mm [14]. Performance tests demonstrated the intrinsic resolution of the scanner to be 4.6–5.7 mm in the transaxial direction and 4.0–5.3 mm in the axial direction. Patients were restrained to the scanner bed using a head holder. A 10 min transmission scan was performed with a ^68^Ge/^68^Ga rod source for attenuation correction before the emission scan. PET data were reconstructed using a Hanning filter with smoothing of 5.0 mm full-width at half-maximum in the transaxial direction [15]. A small cannula was inserted into the right brachial artery for blood sampling before tracer administration. A venous line was obtained from the left antecubital vein for the ^15^O-water injection.

Each patient inhaled a single dose of C^15^O, approximately 1000 MBq, from a nasal tube for CBV measurement, and a static PET scan was started about 50 s after C^15^O inhalation for 3 min. Arterial blood was sampled twice during the static C^15^O scan to calculate a CBV image (Figure 1). Ten minutes after the C^15^O scan, a 3-min PET acquisition was started with a slow bolus inhalation of 1500 MBq/min ^15^O_2_ for 30 s to obtain images of oxygen consumption (OEF and CMRO_2_) using the autoradiography (ARG) method [16,17] (Figure 1). The radioactivity of the blood samples was immediately measured with an automatic arterial blood sampling system (ABSS) consisted of a scintillation counter (Apollome Co. Ltd., Kobe, Japan) and a mini pump (AC-2120, Atto Co., Tokyo, Japan). Arterial blood was sampled at a constant rate of 7 mL/min for the first 2 min by the ABSS, followed by manual sampling of 0.5 mL blood every 20 s during the remaining scan time [18]. Radioactivity counted by the ABSS was calibrated with that of the arterial blood sampled manually. The decay of the radioactivity from dynamic PET and blood data was corrected to the starting point of each scan. The dispersion of the external tube in the arterial curves was corrected with a double exponential dispersion function [18]. Total O_2_ content in the arterial blood was measured from one blood sample obtained manually to calculate CMRO_2_. An arterial blood gas test was also performed using the same blood sample to determine the partial pressure of CO_2_ (PaCO_2_), O_2_ (PaO_2_), pH, and hematocrit [17]. For CBF measurement, a dynamic PET scan was started at the time of the bolus injection of 555 MBq H_2_^15^O from the venous line with frame durations of 30 × 2 s, 6 × 10 s, and 3 × 20 s (39 frames in total) [13]. Acetazolamide (1.0 g/60 kg BW, maximum dose of 1.0 g) in 10 mL saline was administered from the venous line for more than 1 min. The second ^15^O-water PET scan was started 10 min after ACZ administration [17,19]. The scan protocol was the same as the first ^15^O-water scan.

### 2.3. Calculation of Hemodynamic PET Images

The CBF image was calculated based on a 1-TCM expressed by the following equation.
*M(t) = K*_1_*C_a_(t) ⊗ e *^−*k*2*t*^ + *V_0_Ca(t)*, (1)
where *M*(*t*) is radioactivity concentrations in the brain tissue obtained from PET images and *C*_a_(*t*) is the arterial input measured from sampled blood [12]. *K*_1_ and *k*_2_ are the rate constants for influx and outflux of the tracer, V_0_ expresses vascular volume of the arterioles to arterial capillaries, and ⊗ indicates the operation of convolution. Equation (1) can be modified as the following equation using three different *w**_i_* values (*i* = 1–3) as weights [13]:(2)$$\int_{0}^{T}w_{i}\left(t\right)M\left(t\right)dt=K_{1}\int_{0}^{T}w_{i}\left(t\right)C_{a}\left(t\right)⊗e{\;}^{-k_{2}t}dt+V_{0}\int_{0}^{T}w_{i}\left(t\right)C_{a}\left(t\right)dt\;,$$
(3)$$\frac{\int_{0}^{T}w_{3}\left(t\right)C_{a}\left(t\right)dt\times \int_{0}^{T}w_{1}\left(t\right)M\left(t\right)dt-\int_{0}^{T}w_{1}\left(t\right)C_{a}\left(t\right)dt\times \int_{0}^{T}w_{3}\left(t\right)M\left(t\right)dt_{\;}}{\int_{0}^{T}w_{3}\left(t\right)C_{a}\left(t\right)dt\times \int_{0}^{T}w_{2}\left(t\right)M\left(t\right)dt-\int_{0}^{T}w_{2}\left(t\right)C_{a}\left(t\right)dt\times \int_{0}^{T}w_{3}\left(t\right)M\left(t\right)dt_{\;}}\;\;=\frac{K_{1}\left[\int_{0}^{T}w_{3}\left(t\right)C_{a}\left(t\right)\text{d}t\times \int_{0}^{T}w_{1}\left(t\right)C_{a}\left(t\right)⊗e{\;}^{k_{2}t}\text{d}t-\int_{0}^{T}w_{1}\left(t\right)C_{a}\left(t\right)\text{d}t\times \int_{0}^{T}w_{3}\left(t\right)C_{a}\left(t\right)⊗e{\;}^{k_{2}t}\text{d}t_{\;}\right]}{K_{1}\left[\int_{0}^{T}w_{3}\left(t\right)C_{a}\left(t\right)\text{d}t\times \int_{0}^{T}w_{2}\left(t\right)C_{a}\left(t\right)⊗e{\;}^{k_{2}t}\text{d}t-\int_{0}^{T}w_{2}\left(t\right)C_{a}\left(t\right)\text{d}t\times \int_{0}^{T}w_{3}\left(t\right)C_{a}\left(t\right)⊗e{\;}^{k_{2}t}\text{d}t_{\;}\right]},$$
where we used *w*_1_ (*t*) = 1, *w*_2_ (*t*) = *t* and *w*_3_ (*t*) = $\sqrt{t}$ as weighting functions. Equation (3) can be solved for *K*_1_ using the lookup table with variable *k*_2_, brain tissue activity from PET data, and *C*_a_ from blood samples. Finally, the V_0_ value is obtained from *K*_1_ and *k*_2_ using Equation (2). Details of the weighted integral method for ^15^O-PET studies are described elsewhere [12,13]. Since regional CBF and V_0_ values are affected by the delay (an arterial arrival time), we applied the slope method for pixel-by-pixel delay correction to reduce estimation errors [13,20], where a slope of each brain pixel time-activity curve (TAC) was adjusted to the arterial TAC from ABSS (Figure 1). This method also provided a delay image from estimation of the arterial arrival time [13]. A fixed dispersion (4 s) for peripheral arteries was used in CBF calculation [12,21]. CMRO_2_ and OEF images were also calculated by the ARG three-step method using the image data of the baseline ^15^O-water, C^15^O and ^15^O_2_ scan (Figure 1) [16].

### 2.4. Image Analysis

Regional values were obtained using multiple circular regions of interest (ROIs) placed bilaterally on the cortical territories of the MCA (Figure 2). A total of 30 ROIs with a fixed size of 10 mm in diameter were put on each hemisphere at several slice levels using a co-registered individual 3D-T1WI MRI so as to avoid areas of infarction, if any. The same ROIs were transferred to all parametric images (CBF, OEF, CMRO_2_, CBV, V_0_, and delay) of each subject, and the 30 ROI values were averaged for each hemisphere [17,22]. Cerebrovascular reactivity (CVR) was defined as % change in CBF using the equation [(post-ACZ CBF) − (baseline CBF)]/(baseline CBF) × 100%. To estimate regional arterial perfusion pressure (App), a CBF/V_0_ ratio (/min) was calculated using regional CBF and V_0_ ROI mean values [11,23].

### 2.5. Statistical Analysis

Hemodynamic parameters for each group were compared between the two hemispheres using repeated-measures analysis of variance (ANOVA) and a post-hoc paired *t*-test. The two conditions before and after ACZ administration were also compared for CBF and V_0_ in each hemisphere using the same test. Hemispheric differences between the unilateral SOL and MMD groups were compared using a one-way ANOVA with a post-hoc Tukey test. The relationships between the two parameters, such as arterial delay vs. CVR and CBF/V_0_ vs. CBF, were assessed and a Pearson’s correlation coefficient was obtained. The relationships between changes in CBF/V_0_ (ΔCBF/V_0_) and OEF were also observed. The ratio of patients with past neurologic events were evaluated between SOL and MMD using a χ^2^ square test. ΔCBF/V_0_ of patients with or without this past history was also compared using a *t*-test. We considered *p* values less than 0.05 statistically significant.

## 3. Results

Table 2 shows the mean ± standard deviation (SD) of BP, PaCO_2_, and cerebral hemodynamic parameters including CBF and V_0_, after ACZ administration for each group. The values of SOL group were divided into ipsilateral and contralateral hemispheres, and the MMD group were classified as ipsilateral, contralateral, or bilateral hemispheres, depending on the affected side.

In the SOL group, all baseline parameters were significantly different in the ipsilateral hemisphere compared with the unaffected contralateral hemisphere, with a lower CBF, V_0_, (*p* < 0.0001), and CMRO_2_ (*p* < 0.01), and a higher CBV (*p* < 0.05) and OEF (*p* < 0.01). The delay at baseline was significantly greater in the ipsilateral hemisphere than in the contralateral hemisphere (*p* < 0.0001) and it did not change after ACZ administration, although some patients showed a slight decrease (see Figure 6a). CVR values were significantly different between the hemispheres (*p* < 0.0001). All hemispheres showed significant increases in CBF after ACZ administration (*p* < 0.0001), but not in V_0_.

In the MMD group, baseline V_0_ and CMRO_2_ were significantly lower, and OEF and delay were significantly higher in bilaterally impaired hemispheres among the three groups (*p* < 0.05). Unilateral MMD patients showed significantly greater CBV in the affected hemisphere compared with the less affected contralateral hemisphere (*p* < 0.05). Similar to SOL patients, CBF was increased significantly in all hemispheres of MMD patients after ACZ administration, while V_0_ was not changed. Bilaterally impaired MMD patients showed significantly lower post-ACZ CBF and V_0_ than patients with unilateral MMD (*p* < 0.05).

In comparison between the SOL and MMD groups, CBV and CMRO_2_ showed greater values in MMD than in SOL. Even in the contralateral hemisphere, MMD tended to show higher values compared with the hemispheres of the SOL group. The bilateral MMD group showed significantly greater hemispheric OEF and delay compared with the contralateral hemisphere of SOL (*p* < 0.01). CBF/V_0_ and ΔCBF/V_0_ did not show differences between hemispheres, probably because of large SDs.

The relationship between delay and CVR in all patients is demonstrated in Figure 3. The plots show significant negative correlations in all hemispheres (r = −0.32, *p* < 0.005) and in the SOL group (r = −0.39, *p* < 0.01), but not in the MMD group (r = −0.14, *p* = 0.41). Delay time was also correlated well with V_0_ (r = −0.64, *p* < 0.00001 in total) for both SOL (r = −0.75, *p* < 0.00001) and MMD (r = −0.43, *p* < 0.001). These results indicate that impaired vascular reactivity in small vessels is related to the delay of arterial blood but not to OEF elevation (gray plots: 51.5% is the upper limit of our normal OEF). Figure 4 shows changes in baseline CBF as a function of CBF/V_0_ for all hemispheres. Linear and logarithmic regression lines showed a similar correlation of r = 0.46 (*p* < 0.0001) for both analyses, but the root mean square error (RMSE) for the latter was smaller than the former regression (15.6 vs. 6.9).

Changes in CBF/V_0_ before and after ACZ administration (ΔCBF/V_0_) were weakly correlated with OEF (Figure 5, r = −0.23, *p* = 0.04); however, the reduction in CBF/V_0_ after ACZ administration tended to correlate with past neurologic events such as TIA before the PET scans, although OEF elevation did not show relevance. This tendency was greater in the SOL group than in the MMD group (16/48 hemispheres vs. 7/36 hemispheres, respectively, *p* = 0.07, χ^2^ test). ΔCBF/V_0_ values were significantly lower in symptomatic patients (−11.1 ± 10.7/min) compared with asymptomatic patients (5.4 ± 13.5/min) (*p* < 0.00001, *t*-test).

Figure 6a shows parametric images of a representative case of SOL with right ICA occlusion who had a minor stroke. This patient had reduced CVR in the right MCA territory and showed a slight, but not significant elevation of OEF in the same region. The CBF/V_0_ images showed a significant decrease in the right frontal lobe after ACZ administration. Figure 6b shows images of a patient with bilateral MMD without neurological symptoms. This case showed a slight global CBF reduction, OEF elevation, and poor CVR (less than 10%) in the bilateral hemisphere. Delay and CBF/V_0_ images did not show significant changes after ACZ administration.

## 4. Discussion

In the present study, we investigated differences in cerebral circulatory hemodynamics between two cerebrovascular disorders, the SOL group and the MMD group, using PET parameters of CBF, blood volumes of CBV and V_0_, and regional CBF/V_0_ which is supposed to reflect the App. CVR was evaluated by an ACZ challenge test and CVRs were also compared between the SOL and MMD groups. CBV and CMRO_2_ were significantly different between the SOL and MMD groups. Baseline V_0_, OEF and delay showed significant differences between the contralateral hemisphere of SOL and the hemispheres of bilateral MMD. The ipsilateral hemisphere of unilateral MMD patients showed a similar delay to the contralateral hemisphere, which was slightly longer than the unaffected side of SOL patients. The delay was correlated with CVR in the SOL group (r = −0.39, *p* = 0.01), but not in the MMD group (r = −0.14, *p* = 0.41) (Figure 3). The MMD group showed greater venous volume calculated by CBV—V_0_ because CBV values were greater and V_0_ values were not different or smaller in the hemispheres of MMD. Figure 6 showed almost no changes in delay and CBF/V_0_ images in MMD, while global delay time decreased slightly in SOL. These findings indicate that MMD may cause hemodynamic impairment in both hemispheres even in patients with unilateral MMD, suggesting the process of hemodynamic change in MMD may be different from SOL patients.

MMD is usually caused by progressive arterial steno-occlusive changes in the intracranial ICA, particularly in the Willis ring; however, the hemodynamic status is considered to be different from unilateral SOL patients. CBV is usually increased in MMD, not only in the regions of moyamoya vessels, but also in other cerebral cortices [24], which was also observed in the present study as the difference between SOL and MMD. Since V_0_, the volume of the arterioles to the arterial parts of capillaries, did not differ in the regions with CBV elevation (Table 2), the significant CBV increase in MMD may be caused by inactive capillaries with less reactivity and the venous parts of the vessels. Chronic changes in perfusion pressure reduction in both ipsilateral SOL and MMD may cause a decrease in the functional arterial part represented by V_0_, and an increase in inactive vascular volume. Delay time was correlated with CVR in the SOL group, indicating that vasodilatory changes in functioning arteries caused a decrease in arterial velocity. Only SOL showed this correlation, suggesting different cerebral hemodynamics between MMD and unilateral SOL, especially in the functional arterial part of the vessels.

The reduction in CBF/V_0_ after ACZ administration was correlated with a past history of neurological symptoms, including TIA (Figure 5). ΔCBF/V_0_ was significantly lower in symptomatic patients than in asymptomatic patients, although OEF elevation showed no correlation with symptoms. Thus, the impairment of vascular reactivity was considered to be related to TIA and other neurological symptoms. This tendency was observed in the SOL group about twice as much as it was in the MMD group (16/48 vs. 7/36, *p* = 0.07). Development of complicated collateral circulation in MMD patients during chronic circular insufficiency may permit them to resist the transient regional CBF/V_0_ reduction after ACZ administration. Baseline V_0_, assumed to be the functional arterial volume, was not different between symptomatic and asymptomatic patient groups (1.24 ± 0.38 and 1.39 ± 0.51, respectively), suggesting that the reduction of baseline V_0_ alone would not be an index for the prediction of neurological events. On the other hand, decreased CBF/V_0_ may be a risk factor for patients with SOL and would be beneficial information before selecting a treatment.

The correlation between CBF/V_0_ and CBF was observed when comparing all hemispheres (Figure 4), and reflects theoretical hemodynamic changes in regional cerebral circulation based on physiological autoregulation. This relationship was also reported in a previous study [13]. The assumption of cerebral hemodynamic stages by Powers et al. is based on the physiological theory of CBF autoregulation, which can be applied to patients with common SOL and some MMD patients [7,19,22]. The result of CBF/V_0_ vs. CBF correlation in the present study corresponds well with the theory of autoregulation. However, the relationship between vascular functions and oxygen consumption does not necessarily show complete agreement. Several previous studies showed that the conditions of OEF elevation and CVR reduction were not necessarily identical even in SOL patients [17,25,26]. In this study, bilateral MMD showed significantly higher OEF compared with the contralateral hemisphere of SOL, although CVR was not significantly different. The common assumption of hemodynamics may need to be modified in MMD with different processes and conditions of cerebral circulation compared with unilateral SOL.

In the present study, patients’ past neurological events were related to the reduction in CBF/V_0_ after ACZ administration, rather than elevation of OEF values. This result indicates that OEF and ΔCBF/V_0_ are biomarkers representing different physiologic changes in the CVD patients [17]. Both hemodynamic parameters would be risk factors for recurrent neurological events such as infarction [5,17,22,27]. The parameter of CBF/V_0_ may not be stable because V_0_ is significantly influenced by the delay time in the arterial tracer arrival [12,13]. To avoid estimation errors of the delay time in CVD patients, we proposed the pixel-by-pixel calculation method with delay correction [13]. Regional CVR was correlated with delay, which is another evidence of a close relationship between the volume or density of functional arteries and arterial velocity as mentioned above. The reduction in cerebral perfusion pressure would induce a regional vasodilatory change, however, the functional arteries may not necessarily be increased in the chronic phase of CVD, as we observed in ipsilateral V_0_. Our previous studies also supported the same laterality [13,23,25]. Although regional CBF/V_0_ values in MMD patients may be more unstable than in SOL patients because of their complicated collateral circulation, evaluation using average values of greater area, like vascular territories or hemispheres, would be able to estimate dynamic hemodynamic changes.

This study has several limitations including the limited number of patients studied and the different mean age and age range of the two groups. The frame time (2 s) of the initial phase in dynamic ^15^O-water PET may not have been small enough when considering the influence of estimation errors of delay on CBF and V_0_ values [12,13]. However, lower radioactivity in brain PET due to a shorter frame time would also induce substantial errors with a large amount of noise. Because the differences in delay between the hemispheres are about 0.2 s or smaller, regional V_0_ around 1.5 (mL/100 g) may be shifted by 10–20% from the actual volume according to our simulation in the previous study [13]. Therefore, this inter-hemispheric laterality of V_0_ would be maintained even after precise error correction. Recent PET detectors with improved sensitivity may allow a better time resolution compared with the old scanner used in this study. If we apply a PET/MRI scanner and time-to-peak MR image for PET delay estimation, more precise V_0_ images may be available. Further studies using a new scanner and a greater number of CVD patients are expected to confirm our findings.

## 5. Conclusions

In conclusion, the hemodynamic status of cerebral circulation may vary according to the chronic process of steno-occlusive change and the development of collateral circulation. MMD patients showed greater inactive vascular volume compared with common atherosclerotic SOL patients. ^15^O-PET with an ACZ challenge test, a useful tool to evaluate physiologic changes, may elucidate hemodynamic differences between the two diseases.

## Figures and Tables

**Figure 1 diagnostics-11-01820-f001:**
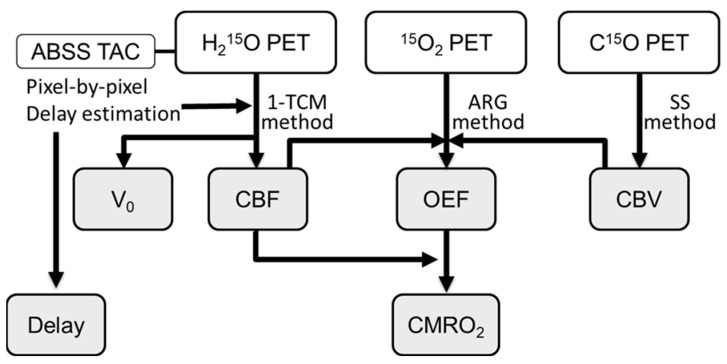
Flow chart of PET scans and parametric image (gray panels) calculation. The SS method for C^15^O scan means the steady state calculation method after the slow bolus C^15^O inhalation. Other abbreviations are defined in the main text.

**Figure 2 diagnostics-11-01820-f002:**
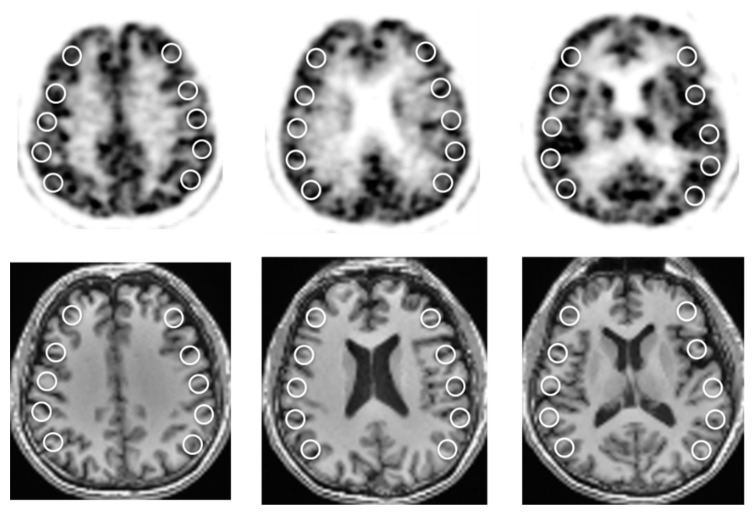
Multiple ROIs placed on bilateral MCA territories of CBF brain slices (**upper**). A total of thirty ROIs with a fixed size of 10 mm in diameter were placed at several slice levels using co-registered T1WI-MRI (**bottom**). The same ROIs were applied to all PET images.

**Figure 3 diagnostics-11-01820-f003:**
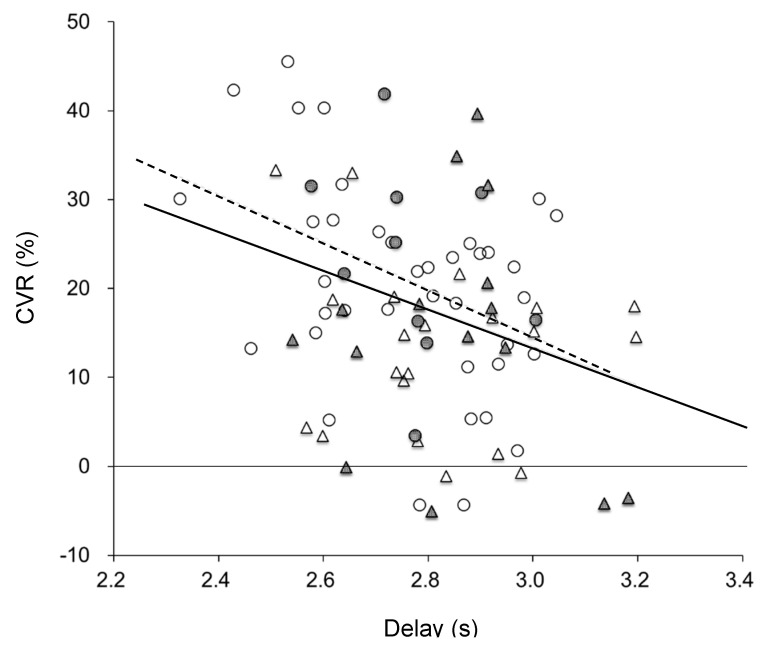
Relationships between baseline delay and CVR (% change in CBF) after the ACZ injection in CVD patients with SOL (circles) and MMD (triangles). Delay and CVR showed a negative correlation in all patients (solid line: *y* = −22*x* + 78, r = −0.32, *p* = 0.003), as well as in the SOL group (dashed line: *y* = −27*x* + 94, r = −0.39, *p* = 0.006), while patients with MMD did not show a significant correlation (r = −0.14, *p* = 0.41). Gray marks indicate the cerebral hemispheres with OEF elevation (i.e., misery perfusion).

**Figure 4 diagnostics-11-01820-f004:**
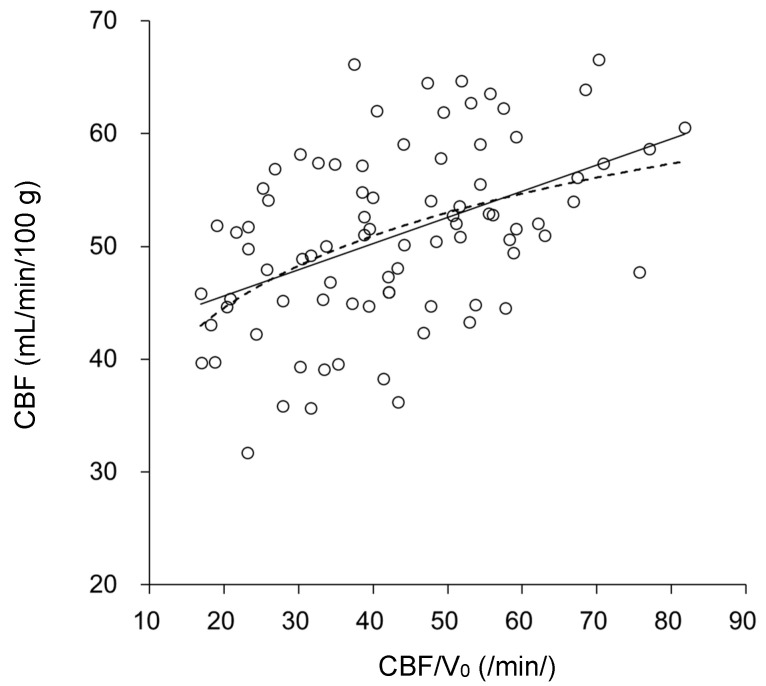
Relationship between CBF/V_0_, related to cerebral App, and baseline CBF for all hemispheres of SOL and MMD patients. In the lower CBF/V_0_ range, CBF tended to be decreased as a function of CBF/V_0_. The solid line shows a linear regression (*y* = 0.23*x* + 41.0, r = 0.46, *p* < 0.0001, RSME = 15.9) and the dashed line shows a logarithmic regression (*y* = 9.2 ln(*x*) + 16.9, r = 0.46, *p* < 0.0001, RSME = 6.9) which tends to plateau at greater CBF/V_0_ levels.

**Figure 5 diagnostics-11-01820-f005:**
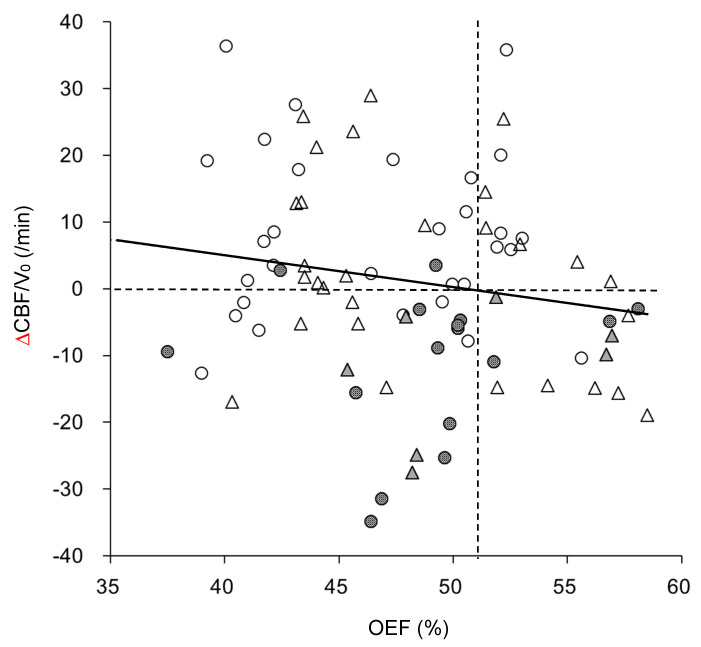
Relationship between OEF and changes in CBF/V_0_ (ΔCBF/V_0_) after ACZ administration. Circles are hemispheres of SOL and triangles represent MMD patients. Gray symbols indicate responsible hemispheres with a past history of TIA or other neurological symptoms. The graph shows that most of the symptomatic patients had negative ΔCBF/V_0_ and the ratio of negative ΔCBF/V_0_ was greater in the SOL group than in the MMD group. The vertical dashed line shows the upper limit of normal OEF in our institute. The solid line indicates the linear regression for all plots, which shows a weak correlation (r = −0.23, *p* = 0.04, *y* = −0.61*x* + 30.3).

**Figure 6 diagnostics-11-01820-f006:**
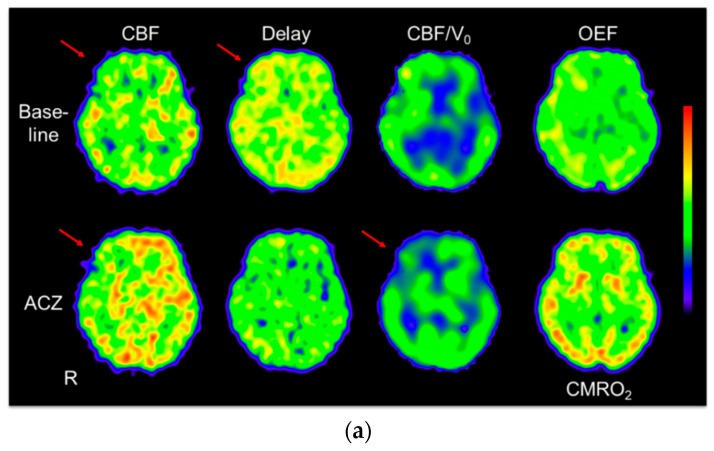
(**a**) Images of a representative case from the SOL group who had neurological events before the PET study. The patient had an occlusive lesion in the right ICA and suffered from white matter lacunar infarction. Baseline CBF showed a slight decrease in the right frontal lobe (red arrow) and reduced CVR in the right middle cerebral artery (MCA) territory. The baseline delay image showed slightly longer delay in the right MCA area than in the left side (red arrow), and this laterality disappeared after acetazolamide (ACZ) administration. The CBF/V_0_ images also showed a significant decrease in the right frontal lobe after ACZ administration (red arrow) despite no change in other brain regions. The OEF image showed slight elevation in the right MCA territory compared with the left side and the CMRO_2_ image had no laterality. (**b**) Images from a case of bilateral MMD without neurological symptoms. She underwent O-15 PET and MRI because her sister had MMD. Cortical CBF slightly reduced globally, which induced slight OEF elevation in the bilateral hemisphere. After ACZ administration, delay and CBF/V_0_ images did not show significant changes, but CVR was less than 10%. R: right side of the brain, the upper and lower limits of the color window are CBF: 0–70 mL/min/100 g, Delay: 0–4.5 s, CBF/V_0_: 0–45/min, OEF: 0–95%, CMRO_2_: 0–5 mL/min/100 g.

**Table 1 diagnostics-11-01820-t001:** Demographic characteristics of patients.

Subject	SOL	MMD
Number	24	18
Age [years, mean ± SD] (range)	66.6 ± 11.3 (37–84)	33.1 ± 16.2 ^†^ (12–70)
Sex		
Male	20	7
Female	4	11
Stroke	11	10
Lesion side		
Right	10	6
Left	14	6
Bilateral	0	6
Steno-occlusion		
ICA (stenosis/occlusion)	17 (9/8)	
MCA (stenosis/occlusion)	7 (3/4)	

SOL: patients with steno-occlusive lesions, MMD: patients with moyamoya disease, ICA: internal carotid artery, MCA: middle cerebral artery, ^†^
*p* < 0.00001 (*t*-test).

**Table 2 diagnostics-11-01820-t002:** Physiological and hemodynamic parameters in ^15^O-PET.

	SOL (*n* = 24)		MMD (*n* = 18)		
	Ipsilateral	Contralateral	Ipsilateral (*n* = 12)	Contralateral (*n* = 12)	Bilateral (*n* = 6 × 2)
BP (mmHg)					
Baseline	91.9 ± 9.8		91.9 ± 11.8		
Post-ACZ	91.8 ± 11.3		94.1 ± 13.3		
PaCO_2_ (mmHg)					
Baseline	40.5 ± 4.4		39.1 ± 3.6		
Post-ACZ	39.5 ± 4.1		38.0 ± 3.5		
CBF (mL/min/100 g)					
Baseline	47.9 ± 7.4 ^c,^*	53.0 ± 6.0	51.8 ± 8.5	55.4 ± 7.9 *	47.8 ± 8.9
Post-ACZ	55.3 ± 8.6 ^a,c,^*	66.3 ± 6.1 ^a,^*^,†^	57.6 ± 10.0 ^b,^*	65.1 ± 8.6 ^a,^*	54.0 ± 12.5 ^b,g,†^
CVR (% Change)	16.2 ± 12.4 ^c^	25.7 ± 8.7	11.5 ± 11.9	18.1 ± 11.0	13.6 ± 12.7
V_0_ (mL/100 g)					
Baseline	1.26 ± 0.48 ^d^	1.59 ± 0.56 *	1.21 ± 0.46	1.36 ± 0.43	1.10 ± 0.37 ^g,^*
Post-ACZ	1.30 ± 0.37 ^c^	1.93 ± 0.57 ^†^	1.50 ± 0.74	1.75 ± 0.84	1.14 ± 0.15 ^g,†^
CBV (mL/100 g)	3.91 ± 0.72 ^e,†^	3.62 ± 0.63 ^†,^*	5.23 ± 1.67 ^g,†^	4.77 ± 1.07 ^†,^*	5.79 ± 1.71 ^g,†^
CMRO_2_ (mL/min/100 g)	2.86 ± 0.35 ^d,†^	3.00 ± 0.27 *	3.32 ± 0.37 ^†^	3.40 ± 0.28 ^†,^*	2.96 ± 0.49 ^f^
OEF (%)	49.0 ± 16.0 ^d^	45.2 ± 5.8 ^†^	48.7 ± 5.9	46.2 ± 2.7	52.5 ± 5.2 ^g,†^
Delay (s)	2.82 ± 0.15 ^c^	2.68 ± 0.18 ^†^	2.78 ± 0.14	2.77 ± 0.14	2.96 ± 0.21 ^f,†^
CBF/V_0_ (/min)	41.8 ± 12.3	39.7 ± 10.3	42.1 ± 13.3	42.2 ± 13.5	45.0 ± 13.0
ΔCBF/V_0_(/min)	−0.63 ± 12.3	0.28 ± 15.2	−1.45 ± 16.2	0.07 ± 0.14	−1.48 ± 11.3

BP, mean blood pressure; PaCO_2_, arterial CO_2_ partial pressure; CVR, cerebrovascular reactivity; ACZ, acetazolamide; ΔCBF/V_0_, changes in CBF/V_0_ after ACZ administration; ^a^
*p* < 0.0001, ^b^
*p* < 0.05, comparing conditions before and after ACZ injection; ^c^
*p* < 0.0001, ^d^
*p* < 0.01, ^e^
*p* < 0.05, comparing two hemispheres in SOL patients (repeated-measures ANOVA with post-hoc test); ^f^
*p* < 0.01, ^g^
*p* < 0.05, comparison among three groups of MMD hemispheres (ANOVA with post-hoc Tukey test); ^†^
*p* < 0.01, * *p* < 0.05, comparison between hemispheres of SOL vs. MMD (ANOVA with post-hoc Tukey test).

## Data Availability

Data available on request due to restrictions eg privacy or ethical.

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
