# Peer review of "Differences in Hemodynamic Alteration between Atherosclerotic Occlusive Lesions and Moyamoya Disease: A Quantitative 15O-PET Study"

_diagnostics, 2021, doi:10.3390/diagnostics11101820_

Round 1

Reviewer 1 Report

In this manuscript, by using 15O-PET the authors investigated hemodynamic status in patients with atherosclerotic stenoocclusive lesions (SOL) and moyamoaya disease (MMD), and attempted to elucidate differences between the two pathological conditions. The most important parameter in this study that is not obtainable in other studies is V0, which is an arterial blood volume based on 15O water CBF study. The main findings on V0 are 1) baseline CBV (total blood volume) is larger in the affected than in unaffected hemispheres, but V0 is smaller, 2) ACZ administration shows no significant increases in V0 for both hemispheres, which contrast to significant increases in CBF, and 3) patients with SOL and MMD have different profiles of CBV and V0, with greater inactive vascular and venous volumes in MMD. Based on the findings the authors conclude that 15O-PET including an acetazolamide challenge test is useful to characterize differences between the two diseases.

Overall, the data are clearly presented and the manuscript is easy to follow. The numbers of subjects were 24 for SOL and 18 for MMD, which are not sufficiently large as mentioned in the limitations section, but at the same time seem to be reasonable given the limited availability and technical difficulty of PET applying both 15O-labeled gas and water (resting state and ACZ). The full analysis of various hemodynamic parameters, which include “classical” 15O PET parameters (CBF, CBV, OEF, CMRO2), V0, Delay, and ACZ responses, is of significant important and interest to readers in this field even though a technical concern on V0.

Major comments:

I have a concern about the accuracy and reliability of Delay and hence V0, determined form the dynamic analysis of 15O water PET data. The authors used the method developed in Reference [12], in which they actually evaluated the estimated errors in V0 induced by Delay time shift, but the delay error itself is unknown. The frame time of the current study was 2 secs in early phase, and the accuracy of Delay is presumably over hundreds of msec, corresponding to over 50% errors in V0. In Table 2, differences in Delay between hemispheres are <200 msec and V0 values are smaller in the ipsilateral hemisphere; however, if true Delay in the ipsilateral hemisphere is longer than the present results, even with hundreds of msec, V0 should be reversed. This issue was already mentioned in the limitations section. However, given that the discussions in the study depend strongly on V0 and its great impact, further explanation is needed.

Comments relating this issue are as below.

Page 5: “In the SOL group, all baseline parameters were significantly different in the ipsilateral hemisphere compared with the unaffected contralateral hemisphere, with a lower CBF, V0, (p < 0.0001)”

Page 11: “Reduction of cerebral perfusion pressure would induce regional vasodilatory change, however, the functional arteries may not necessarily be increased in chronic phase of CVD as we observed in ipsilateral V0.”

Lower V0 in the ipsilateral hemispheres compared to the contralateral is interesting, important finding. Are there references supporting the current observation?

Page 11: “Regional V0 was correlated with delay, which is another evidence of close relationship between the volume of functional arteries and arterial velocity.”

The correlation between V0 and Delay could also be explained by the methodologic correlation induced by the estimation errors in Delay.

In Table 2, Delay values are provided in single row. If I understand it, Delay values in 15O water PET were determined separately with rest and ACZ exams. Are there significant differences?

Minor comments:

Page 4: “Details of the weighted integral method for 15O-PET studies are described elsewhere [11,12].”

Types of weighting function used in the analysis are of quite importance as determinant of Delay accuracy, but not found in Refs 11 and 12.

Figure 1: “Flow chart of PET scans and parametric image (gray panels) calculation.”

Is Delay incorporated also in the calculation of O2 image (OEF calculation)?

Page 7: “These results indicate that impaired vascular reactivity in small vessels is related with the delay of arterial blood but not with OEF elevation (gray plots).”

Definition of OEF elevation, or misery perfusion, should be provided in the text.

Page 8 and Figure 6(a): “The CBF/V0 images also showed a significant decrease in the left frontal lobe after ACZ administration (red arrow)”

Right frontal?

Author Response

We wish to express our appreciation to the reviewer for the insightful comments and suggestions which have helped us significantly improve the paper.

Major comments:

Q: I have a concern about the accuracy and reliability of Delay and hence V0, determined form the dynamic analysis of 15O water PET data. The authors used the method developed in Reference [12], in which they actually evaluated the estimated errors in V0 induced by Delay time shift, but the delay error itself is unknown. The frame time of the current study was 2 secs in early phase, and the accuracy of Delay is presumably over hundreds of msec, corresponding to over 50% errors in V0. In Table 2, differences in Delay between hemispheres are <200 msec and V0 values are smaller in the ipsilateral hemisphere; however, if true Delay in the ipsilateral hemisphere is longer than the present results, even with hundreds of msec, V0 should be reversed. This issue was already mentioned in the limitations section. However, given that the discussions in the study depend strongly on V0 and its great impact, further explanation is needed.

A: As pointed out by the reviewer, errors in delay time estimation may induce substantial over- or under-estimation of calculation results especially for V0 values. According to our simulation in the previous study (ref. #12), regional V0 around 1.5 (mL/100g) may be shifted about 80% by a 1.0 sec (= 1,000 msec) error of delay estimation, but it would be reduced to about 10-20% shift by an estimation error of 200 msec or less. Therefore, this laterality of smaller V0 in the ipsilateral hemisphere than in the contralateral hemisphere will be maintained even after the precise error correction. But we agree with the reviewer that this is an important discussion and we explained this issue in Discussion (Page 12).

Q: Comments relating this issue are as below.

Page 5: “In the SOL group, all baseline parameters were significantly different in the ipsilateral hemisphere compared with the unaffected contralateral hemisphere, with a lower CBF, V0, (p < 0.0001)”

Page 11: “Reduction of cerebral perfusion pressure would induce regional vasodilatory change, however, the functional arteries may not necessarily be increased in chronic phase of CVD as we observed in ipsilateral V0.”

Lower V0 in the ipsilateral hemispheres compared to the contralateral is interesting, important finding. Are there references supporting the current observation?

A: We thank the reviewer for this valuable comment and interest in our study. Our previous studies also showed similar results with lower V0 values in the ipsilateral than in the contralateral hemisphere (ref. # 13, 23, 25). The delay time was estimated on a slice-by-slice basis in the two older studies (#23 and 25), and we improved this estimation using pixel-by-pixel delay correction to obtain precise V0 and CBF in our recent study (#13). However, all of these studies showed similar laterality in V0 and CBF, i.e. V0 is smaller in the ipsilateral than in the contralateral side. We added a line in Discussion and cited these papers (Page 11).

Q: Page 11: “Regional V0 was correlated with delay, which is another evidence of close relationship between the volume of functional arteries and arterial velocity.”

The correlation between V0 and Delay could also be explained by the methodologic correlation induced by the estimation errors in Delay.

A: In this sentence, ‘V0’ is a typo of ‘CBR’. We are sorry about this mistake. This sentence means longer delay is closely correlated with a decrease in volume or density of functional arteries. We corrected the typo and modified the sentence (Page 11).

Q: In Table 2, Delay values are provided in single row. If I understand it, Delay values in 15O water PET were determined separately with rest and ACZ exams. Are there significant differences?

A: As the reviewer pointed out, each patient has two delay images before and after ACZ administration. Because regional delay time was almost the same for the two conditions, only the baseline delay is given in Table 2. Only the ipsilateral hemisphere of SOL showed a slight decrease from 2.82 to 2.69 sec, but the difference was not significant. This result is added in the Results section (Page 6).

Minor comments:

Q: Page 4: “Details of the weighted integral method for 15O-PET studies are described elsewhere [11,12].”

Types of weighting function used in the analysis are of quite importance as determinant of Delay accuracy, but not found in Refs 11 and 12.

A: We are sorry about this inconvenience. Ref 12 (#13 in the revised version) cited a paper by Ohta et al. (J Cerebral Blood Flow Metab 1992) and the weighting function is described in this paper. We added the information of three weighting function after Eq.3 (Page 4).

Q: Figure 1: “Flow chart of PET scans and parametric image (gray panels) calculation.”

Is Delay incorporated also in the calculation of O2 image (OEF calculation)?

A: Since OEF image is calculated using the ARG method, CBF and CBV images are used for calculation of OEF image from O2 PET data. Delay images are the result of pixel-by-pixel delay estimation which is used for delay correction in CBF and V0 image calculation.

Q: Page 7: “These results indicate that impaired vascular reactivity in small vessels is related with the delay of arterial blood but not with OEF elevation (gray plots).”

Definition of OEF elevation, or misery perfusion, should be provided in the text.

A: We apologize for the lack of important information. The upper limit of normal OEF in our center is 51.5 %, and this information is now added in the text (Page 7).

Q: Page 8 and Figure 6(a): “The CBF/V0 images also showed a significant decrease in the left frontal lobe after ACZ administration (red arrow)”

Right frontal?

A: We are sorry about the mistake. ‘left’ is now corrected to ‘right’.

Reviewer 2 Report

Interesting and well written study regarding the differences in hemodynamic alteration between atherosclerotic occlusive lesions and moyamoya disease with a quantitative 15O-PET.

Introduction: can be improved, since it has been demonstrated that Moyamoya can be acquired as a post-infectious disease (ie: Trombatore P et al. A Rare case of postinfectious Moyamoya syndrome: case report and review of the literature. World Neurosurgery, 2020, 140, pp. 213–218) even if rare, could be mentioned.

Material and methods and results: no concerns

Discussion: you could add some lines regarding potential clinical utility of this study in the setting of atherosclerotic occlusive lesions, if any.

Conclusions: ok

Author Response

We wish to express our appreciation to the reviewer for the insightful comments and suggestions.

Q: Introduction: can be improved, since it has been demonstrated that Moyamoya can be acquired as a post-infectious disease (ie: Trombatore P et al. A Rare case of postinfectious Moyamoya syndrome: case report and review of the literature. World Neurosurgery, 2020, 140, pp. 213–218) even if rare, could be mentioned.

A: Thank you very much for this valuable information. We modified Introduction according to the reviewer’s suggestion.

Q: Discussion: you could add some lines regarding potential clinical utility of this study in the setting of atherosclerotic occlusive lesions, if any.

A: We thank the reviewer’s suggestion for acknowledging the importance of the parameters used in this study. We added a line in Discussion as follows: ‘On the other hand, decreased CBF/V0 may be a risk factor for patients with SOL and would be beneficial information before selecting a treatment.’ (Page 11)

Round 2

Reviewer 1 Report

The authors have addressed all my questions.